# Improved Protocol to Study Osteoblast and Adipocyte Differentiation Balance

**DOI:** 10.3390/biomedicines11010031

**Published:** 2022-12-22

**Authors:** Ana Alonso-Pérez, María Guillán-Fresco, Eloi Franco-Trepat, Alberto Jorge-Mora, Miriam López-Fagúndez, Andrés Pazos-Pérez, Antía Crespo-Golmar, José R. Caeiro-Rey, Rodolfo Gómez

**Affiliations:** 1Musculoskeletal Pathology Group, Institute IDIS, Santiago University Clinical Hospital, 15706 Santiago de Compostela, Spain; 2Traumatology and Orthopaedics Service, SERGAS, Santiago University Clinical Hospital, 15706 Santiago de Compostela, Spain

**Keywords:** osteoblast, adipocyte, differentiation, mesenchymal stem cell, osteoporosis, protocol

## Abstract

Adipogenesis-osteoblastogenesis balance-rupture is relevant in multiple diseases. Current human mesenchymal stem cells (hMSCs) in vitro differentiation models are expensive, and are hardly reproducible. Their scarcity and variability make an affordable and reliable method to study adipocyte-osteoblast-equilibrium difficult. Moreover, media composition has been inconstant throughout the literature. Our aims were to compare improved differentiation lab-made media with consensus/commercial media, and to identify a cell-line to simultaneously evaluate both MSCs differentiations. Lab-made media were compared with consensus and commercial media in C3H10T1/2 and hMSC, respectively. Lab-made media were tested on aged women primary pre-osteoblast-like cells. To determine the optimum cell line, C3H10T1/2 and hMSC-TERT cells were differentiated to both cell fates. Differentiation processes were evaluated by adipocytic and osteoblastic gene-markers expression and staining. Lab-made media significantly increased consensus medium induction and overcame commercial media in hMSCs differentiation to adipocytes and osteoblasts. Pre-osteoblast-like cells only properly differentiate to adipocyte. Lab-made media promoted adipocyte gene-markers expression in C3H10T1/2 and hMSC-TERT, and osteoblast gene-markers in C3H10T1/2. Oil Red O and Alizarin Red staining supported these findings. Optimized lab-made media were better at differentiating MSCs compared to consensus/commercial media, and evidenced the adipogenic commitment of pre-osteoblast-like cells from aged-women. C3H10T1/2 is an optimum MSC line by which to study adipocyte-osteoblast differentiation balance.

## 1. Introduction

According to the World Health Organization, musculoskeletal pathologies (MSPs) are the main cause of physical disability in western countries. They affect one in four people older than 20 years old [1]. Even though osteoporosis and osteopenia are MSPs where bone is more affected, these are not the only MSPs with bone alterations [2,3]. In fact, bone alterations are a common denominator in inflammatory MSPs, such as psoriasis [4] or rheumatoid arthritis [3]. Certain MSPs treatments are also associated with bone metabolism disturbances, such as long-term corticoid and proton pump treatments [5]. Nonetheless, bone alterations are also present in MSPs, with more discrete inflammatory components. Indeed, osteoarthritis subchondral bones show evident bone alterations, including osteoporotic and sclerotic areas linked to the evolution of the disease [6].

The rupture of the bone remodeling balance (the relationship between bone formation and bone resorption) results in pathological situations such as osteoporosis and osteopetrosis [7]. Interestingly, osteoblast, the cell type in charge of bone formation, shares a mesenchymal origin with adipocyte. Mesenchymal stem cells (MSCs) have the capacity to differentiate into multiple cell fates. Concretely, osteoblast differentiation and adipocyte differentiation are opposite and strongly related processes. Therefore, the bone formation process depends on the right equilibrium between MSCs differentiation towards osteoblast and adipocyte (osteoblast-adipocyte equilibrium). Consequently, an increase in bone adiposity has been linked with bone loss described in numerous pathologies, such as osteoporosis, osteopenia, obesity, and aging, among others [7,8,9,10].

Today, it is well known that adipogenesis and osteoblastogenesis are competitive and mutually controlled processes [11]. As a result, most pro-adipogenic factors act as osteoblastogenesis inhibitors, and vice versa [12]. Nevertheless, no standardized protocol exists to simultaneously study both processes in vitro in the same cellular context. As has been shown in the literature, the vast majority of experiments have been performed to assess the differentiations made in 3T3-L1 pre-adipocyte cells [13], or MC3T3-E1, SaOS2, or MG-63 pre-osteoblast cells [14]. Even though these cell lines offer proper differentiations, they do not fully resemble the processes from their starting point, MSCs, and they cannot differentiate between both cell fates from a common cell type. Moreover, the composition of proadipogenic and proosteoblastic media varies through the literature, making it difficult to identify the optimum reagent combination to induce these differentiations. This fact limits the evaluation of modulatory factors and novel therapeutic tools acting over the commitment of MSC towards adipogenesis and osteoblastogenesis, as well as during the first steps of the differentiation process. With this in mind, a reliable differentiation protocol and a proper MSCs model that reduces cellular, environmental, and culture conditions variability, offering an interchangeable differentiation process, is required. The aims of this study were to identify proadipogenic and proosteoblastic consensus media in the literature, to compare improved differentiation lab-made media with consensus/commercial media, and to identify a cell-line to simultaneously evaluate both MSCs differentiations. Hence, in this work, we comprehensively analyzed the state-of-the-art osteoblast and adipocyte differentiation protocols and proposed new, consensual, reliable, and more affordable differentiation media by which to acquire these phenotypes. Additionally, we explored the effectiveness of this differentiation media on different MSCs lines in order to obtain an adequate osteoblast-adipocyte differentiation cell model.

## 2. Materials and Methods

### 2.1. Reagents

Dulbecco’s modified Eagle’s medium (DMEM), DMEM supplemented with HAM’S-F12 (DMEM-F12), penicillin/streptomycin, L-glutamine, foetal bovine serum (FBS, lot: BCBW6329), and trypsin were purchased from Sigma-Aldrich (St. Louis, MO, USA). Insulin growth factor-1 (IGF1) was purchased from PeproTech Inc (Cranbury, NJ, USA). Unless referenced, all reagents were purchased from Sigma-Aldrich.

The murine mesenchymal stem cell line C3H10T1/2 was donated by Dr. Pardo (IDIS Institute). Bone marrow human mesenchymal stem cell line hMSC-TERT was donated by Dr. Campana (National University of Singapore, Singapore). Human mesenchymal stem cells (hMSCs) were purchased from Thermo Fisher Scientific Inc. (Waltham, MA, USA).

### 2.2. Literature Revision

To determine the consensual protocol of adipogenic medium, the term “adipogenesis protocol” was searched for in the PubMed database. After a selection of representative paper samples (10%; PubMed best match), a consensus adipogenic differentiation media was established. Moreover, in order to determine IGF1 addition into adipocyte differentiation medium, the term “adipogenesis IGF1” was searched for in the PubMed database. With no date limitation, all the available papers were reviewed. All the protocols studied in both searches were considered to elaborate on an adipogenic consensus medium.

To establish an osteoblastogenic medium consensual protocol, the term “osteoblastogenesis protocol” was searched for in the PubMed database. All the results obtained were analyzed to determine the optimal reagents for an osteoblastogenic consensus medium.

### 2.3. Cell Culture and Differentiation

Human pre-osteoblast-like cells were obtained from total knee replacement surgery of aged women (between 60 and 75 years old) as previously described in Helfrich, Miep H., Ralston, 2003 [15]. The Ethics Committee for Research at Santiago-Lugo Area approved the protocol (CAEIG-2016/258). Informed consent was obtained from all patients. All processes were performed as described in current guidelines and regulations. These cells were age, sex, and pathology matched. Each replicate corresponded to one patient, and no cell pooling was made.

C3H10T1/2 cells were grown in DMEM high glucose, with 10% FBS, 4 mM L-glutamine, and 100 U/mL penicillin/streptomycin. For their differentiation, 10^4^ cells were seeded per well in a 24-well plate and differentiation media was added 6 h after seeding.

Lab-made AD differentiation medium comprised DMEM, 2 μM rosiglitazone, 20 nM IGF1, 1 μM dexamethasone, 60 μM indomethacin, 0.5 mM IBMX, 10 μg/mL insulin, 4 mM L-glutamine, 100 U/mL penicillin/streptomycin, and 10% FBS. Medium was renewed every two days. To obtain an efficient differentiation, three cycles, each one with 96 h of adipocyte differentiation medium and 72 h with maintenance medium, were performed. Maintenance medium consisted of culture medium supplemented with 10 μg/mL insulin. Vehicle concentrations used to solubilize the adipogenic factors were maintained below 0.1% to avoid toxic effects.

Lab-made OB differentiation medium consisted of DMEM high glucose, 10 nM dexamethasone, 5 mM β-glycerol phosphate, 50 μg/mL ascorbic acid-2-phosphate, 4 mM L-glutamine, 100 U/mL penicillin/streptomycin, and 10% FBS. Medium was renewed every two days.

hMSC-TERT cells were cultured in DMEM, 2 mM L-glutamine, 100 U/mL penicillin/streptomycin, and 10% inactivated FBS. To differentiate hMSC-TERT cells to adipocyte, 5 × 10^3^ cells were seeded per well in a 24-well plate and maintained during 72 h in culture medium in order to achieve an hyperconfluent or confluent starting point, respectively. Then, cells were differentiated to adipocyte with the same procedure referred to for C3H10T1/2. For osteoblast differentiation, 3 × 10^5^ hMSC-TERT cells were seeded, per well, in a 6-well plate. Cells were not seeded in a 24-well plate to avoid detachment during the last days of the differentiation process. The following day, osteoblastogenesis was initiated as described for C3H10T1/2. Bone marrow human mesenchymal stem cells were cultured in MesenPro RS Medium (Thermo Fisher Scientific, Waltham, MA, USA) supplemented with MesenPRO RS Growth Supplement (Thermo Fisher Scientific, Waltham, MA, USA). To induce their differentiation, 45 × 10^3^ cells were seeded per well in a 24-well plate. The following day, two differentiation protocols were used: one according to manufacturer differentiation instructions (StemPro Adipogenesis Differentiation Kit (Themo Fisher Scientific, Waltham, MA, USA) StemPro Osteogenesis Differentiation Kit (Thermo Fisher Scientific, Waltham, MA, USA), referred to as commercial media, and the other one, as indicated above for C3H10T1/2.

Human pre-osteoblast-like cells were cultured in DMEM-F12, 4 mM-glutamine, 200 U/mL penicillin/streptomycin, and 10% FBS. To differentiate them, 45 × 10^3^ cells were seeded per well in a 24-well plate. The next day, adipocyte and osteoblast differentiation media were added as formerly referred to for C3H10T1/2.

Every differentiation process was performed for 21 days, as long as cells remained attached.

### 2.4. Gene Expression Analysis

After differentiation, cells were lysed with Tri-Reagent (Sigma-Aldrich, St. Louis, MO, USA) and RNA was isolated with a E.Z.N.A. total RNA kit I (Omega, Bio-Tek, Inc., Norcross, GA, USA), according to manufacturer’s instructions. The samples employed were those with a 260/280 nm ratio higher than 2 and a 260/230 nm ratio higher than 1.7. A High-Capacity cDNA Reverse Transcription Kit (Applied Biosystems, Life Technologies, Grand Island, NY, USA) was used to obtain cDNA from 50 ng of RNA. mRNA expression levels of the marker genes fatty acid binding protein (FABP4), enhancer-binding protein alpha (CEBPA), perilipin 2 (PLIN2), adiponectin (ADIPOQ), peroxisome proliferator activated receptor gamma (PPARγ), alkaline phosphatase (ALPL), osteopontin (SPP1), osteoactivin (GPNMB), runt-related transcription factor 2 (RUNX2), and bone morphogenic protein 2 (BMP2) were assessed by real-time quantitative PCR using BioRad (Hercules, CA, USA) MasterMix and the primers (Sigma-Aldrich, St. Louis, MO, USA) showed in Table 1. Quality control was established through a Non-Template Control as negative control, and a specificity control with dissociation curve analysis. RT-qPCR data relative quantitation was obtained through the ΔΔCt Comparative Method. The housekeeping gene (HPRT) proved no modulation through differentiation processes for any of the cell types used. Other genes were tested as potential housekeeping genes, such as glyceraldehyde-3-phosphate dehydrogenase (GAPDH), glucuronidase beta (GUSB), and TATA-binding protein (TBP) being HPRT the better option. As evidenced in Figure 1, HPRT showed no modifications during the differentiation in any of the cell lines studied.

### 2.5. Cellular Stainings

Colorimetric techniques, such as Alizarin Red and Oil Red O, were performed on fixed cells. When differentiated to adipocytes, fixation was performed with formalin 4% for 1 h at room temperature, and when differentiated to osteoblast, fixed with 70% cold ethanol for 5 min at −20 °C. To dye lipid droplets in adipocyte differentiation, the cells were washed with distilled water and 60% isopropanol. Once the cells were dry, Oil Red O 21% (*w*/*v*) was added for 10 min. After four washes with distilled water, photographs were acquired. Conversely, to evidence calcium deposits related to mineralisation, the cells were incubated with Alizarin Red (40 mM, pH 4.2) for 25 min, and then washed with distilled water. Photos were acquired after drying.

### 2.6. Statistical Analysis

Data are presented as the mean ± standard error of the mean (SEM) for at least three independent experiments. Significant differences were calculated using a Student’s t test. All statistical analyses were performed using Prism software (GraphPad Software Inc., La Jolla, CA, USA), and *p*  <  0.05 was considered significant.

## 3. Results

### 3.1. Adipogenic Consensus Medium

To study osteoblast and adipocyte differentiation balance in a reliable manner, commercial primary mesenchymal stem cells (MSCs) and media are often used. The alternative to this is the use of cell lines and lab-made media. These media are less expensive and have known compositions, but there is no consensus regarding their specific components and their optimization for different cell lines. Therefore, we aimed to investigate more affordable and reliable media with a known and consensual composition throughout the literature.

Comprehensive analysis of a representative sample of articles (Figure 2A) revealed, for adipogenic media composition, a significant variability in each of its common components. Indeed, 70% of the studied protocols used DMEM medium (with glucose concentration over 3.15 g/L) supplemented with 10% FBS. Nonetheless, the concrete type of DMEM used was not even across different protocols. To induce adipocyte differentiation, the constant components found in the literature were glucocorticoids (dexamethasone), IBMX, and insulin. The most frequent concentrations for these compounds were 1 μM, 0.5 mM, and 10 μg/mL, respectively. Multiple protocols also used more additives to further improve the adipogenic differentiation. Among them, thiazolidinediones (rosiglitazone) or indomethacin were the most frequent additives utilized (20% of the protocols). Regardless of media composition, 50% of the protocols described in the literature used cycles of differentiation and maintenance to induce adipocyte differentiation.

The accumulated variability caused by different reagents, concentrations, additives, cells, and differentiation cycles make it difficult to determine the most convenient media and protocol for proper adipocyte differentiation. In addition, after carefully reading the literature, we observed that certain specific additives, like IGF1, which could be easily incorporated into the culture media to improve the differentiation process [16,17,18], were not being used as a common component.

Thus, in response to the literature analysis, we designed and tested a consensus protocol for adipocyte differentiation that pooled the best reagents and concentrations described in the literature. As shown in Figure 2B, the culture of MSCs cells (C3H10T1/2) for 7 days with the consensus media induced a strong and significant augment of adipogenesis marker genes (PLIN2, ADIPOQ, PPARG). In addition, we explored the supplementation of this media with IGF1, a known adipogenic inducer, that may compensate insulin-diminished sensitivity in early differentiation stages of the differentiation process [19]. Addition of IGF1 to the consensus media (lab-made medium) significantly induced the expression of adipogenesis marker genes (FABP4, PLIN2, ADIPOQ) (Figure 2C).

### 3.2. Osteoblastic Consensus Medium

In contrast to adipogenic medium, thorough analysis of the literature regarding osteoblastogenesis differentiation medium revealed that the majority of media have the same constituents, comprising DMEM, low concentrations of dexamethasone, which has been referred as an osteoinducer when used in these conditions [20]; β-glycerol phosphate, necessary for the mineralization process [21]; and 2-phosphate ascorbic acid, a molecule involved in collagen synthesis, as well as in ALPL action [22]. As a result, we selected this osteoblastogenic consensus media for all of the experiments. Our aim was to identify a model to study adipogenic and osteoblastogenic differentiations balance starting from a common point, diminishing environmental and cellular influences. Thus, to avoid the bias of different glucose concentrations, the osteoblastogenic lab-made medium main component was DMEM high glucose. To note, glucose concentrations have been proven not to affect osteoblastic differentiation [23].

### 3.3. Effect of Lab-Made Media vs. Commercial Media on Primary hMSCs

After improving the composition of the media and by pulling the most frequent media components in the literature, we compared the lab-made media/consensus media (adipogenic (AD) or osteoblastogenic (OB) medium) with commercial media using standard differentiation protocols on commercial primary human MSCs.

After 7 days of differentiation, lab-made AD medium induced hMSC adipogenesis. This was evidenced through the significant augment of FABP4, CEBPA, PLIN2, ADIPOQ, and PPARγ gene expressions. Commercial medium also induced the expression of the same marker genes, even though, due to its variability, only PPARγ and ADIPOQ induced expression were statistically significant compared to day 0 (Figure 3A). Unlike the increased expression of adiponectin by the commercial medium, the differences between lab-made and commercial media were not statistically significant.

hMSC differentiated with lab-made OB medium for 7 days increased the expression of osteoblast marker genes SPP1, GPNMB, and BMP2. This expression was higher than commercial medium-induced expression, suggesting a more efficient differentiation process (Figure 3B). Interestingly, neither the commercial medium nor the lab-made medium were able to modulate the expression of ALP and RUNX2 marker genes.

### 3.4. Evaluation of Osteoblast-Adipocyte Commitment of Primary Aged Osteoblast-like Cells

Osteoblast-adipocyte equilibrium rupture has been associated with several pathologies and physiologic conditions, including ageing [8] and menopause [24], among others. Accordingly, we decided to determine whether the lab-made media studied here let us evaluate the osteoblastic and adipogenic commitment of osteoblast-like cells obtained from the bones of aged osteoarthritic women. As shown in Figure 4A, among the osteoblastogenic marker genes studied, only ALPL expression significantly augmented during the osteoblastic differentiation process, which suggested an impaired differentiation process for these cells. On the contrary, adipogenic marker genes FABP4, CEBPA, ADIPOQ, and PPARG significantly incremented their expression from day 14 of differentiation (Figure 4B).

### 3.5. Differentiation Media Evaluation in MSC Lines

Osteoblastic-adipogenic equilibrium in primary MSCs cultures is the ideal set up. Nonetheless, these cultures are expensive and donor variability often hinders the identification of changes in the osteoblastic-adipogenic balance. Thus, two mesenchymal stem cell lines with a proven capacity to differentiate to both cell fates (mice C3H10T1/2, and human hMSC-TERT) were selected. The differentiation induction ability of the proposed media was studied for 21 days in these cell lines. We used mRNA expression of both cell fates’ marker genes (osteoblast and adipocyte) to confirm whether the differentiation processes of MSCs were successfully accomplished.

In confluent cultures of C3H10T1/2 cells, the adipogenic lab-made medium significantly induced the adipogenic marker genes from day 7, peaking at day 21 (Figure 5A). Consistent with this induction of adipogenic marker genes, Oil Red O staining evidenced the progressive intracellular lipid accumulation in these cells from day 7 to day 21 (Figure 5B,C). As observed in the C3H10T1/2 cells, the effect of the adipogenic lab-made medium on hMSC-TERT cells significantly induced the expression of FABP4, ADIPOQ, and PPARG marker genes, even though PPARG induction was not constant during the differentiation process (Figure 5D). Interestingly, PLIN2 expression significantly diminished at day 21 in this cell line (Figure 5D). According to the elevated expression of the adipogenic marker genes on day 21, Oil Red O staining evidenced a significant augment of intracellular lipid accumulation in hMSC-TERT cells (Figure 5B). Nonetheless, total lipid accumulation, as well as the lipid droplets size, was greater in C3H10T1/2 (Figure 5B).

As in adipogenesis, osteoblastogenesis was studied in both mesenchymal stem cell lines using a lab-made consensus medium. Data obtained revealed that the C3H10T1/2 osteoblast marker genes (ALPL, SPP1, GPNMB, RUNX2) significantly augmented during the entire differentiation process (Figure 6A). Alizarin Red staining revealed a significant increase in cell culture mineralization in this cell line (Figure 6B,C). Nonetheless, this was more evident in hMSC-TERT cells (Figure 6B). Interestingly, a similar pattern was observed in hMSC-TERT marker genes expression, although SPP1 decreased its expression and RUNX2 expression remained almost unaltered during the differentiation process (Figure 6D).

## 4. Discussion

In this work, we aimed to establish a consensus and reliable method by which to study the balance between osteoblast and adipocyte differentiation processes. We developed lab-made MSCs differentiation media with known composition, obtained after pooling the most frequent and active reagents used in the literature for osteoblastogenic and adipogenic differentiation. In addition, we improved adipogenic media composition with IGF-1, a known pro-adipogenic factor. We tested them in C3H10T1/2 cells, in hMSC-TERT cells, and in commercial primary hMSCs. In these cell cultures, we proved that these media were reliable, affordable, and efficient, even more so than commercial ones. We confirmed their usefulness through the determination of the pro-adipogenic commitment of pre-osteoblasts-like cells from aged post-menopausal women.

MSC adipogenic-osteoblastogenic differentiation balance disruption is a common feature across different bone alterations. An unbalance towards adipogenesis has been associated with bone fragility [10]. Conversely, deregulation of osteoblastogenesis has been linked to abnormal bone growth [25]. Nevertheless, there is no standardized or consistent in vitro model to efficiently study this balance. The differentiation media are quite variable in terms of their components and concentrations. Similarly, the majority of cell line models studied cannot differentiate both cell fates because they have a fixed commitment, either to osteoblast or adipocyte fate. Consequently, our aim was to identify the most composition stable media and MSCs cellular models to study each arm of the equilibrium, and the balance itself.

Despite adipogenic differentiation media composition having multiple similarities in the literature [26,27,28], the yield rate strongly varies. Boosted by this limitation, we combined the most common protocols from the literature and added IGF-1 [29]. Surprisingly, despite its well-known adipogenic activity, this stimulus has not been previously used as a regular adipogenic differentiation medium component. Consistent with its adipogenic role, this factor significantly boosted the expression of almost all the adipogenic markers studied, which contributed to improve the yield of the adipogenic differentiation process [30,31]. In contrast to adipogenesis, osteoblastogenesis differentiation media exhibit fewer variable components in the literature [26,32,33]. Although it is well known that certain factors, such as BMPs and others, strongly boost this process [34,35], we did not supplement the osteoblastic media with these factors to avoid fingerprints that could mask any normal change associated with the process. In addition, it has been referred that C3H10T1/2 cells treated with BMP2 differentiate not only into osteoblasts, but also into chondrocytes or adipocytes [36]. Since our aim was to obtain a cell model that allowed us to study osteoblast-adipocyte balance, these kinds of additives could impair proper cell commitment identification. Thus, we developed a lab-made osteoblastic differentiation medium with a consensus composition from across the literature.

To test the lab-made media efficiency, several marker gene expressions were assessed. Regarding adipogenesis, FABP4 is present in mature adipocytes and has been also described as a marker for adipocyte progenitors [37], PLIN2 is known for surrounding lipid droplets; therefore, it is a marker of fat accumulation [38]. ADIPOQ plays an essential role in differentiation from preadipocytes to mature adipocyte has been established [39], and PPARγ is widely used as a marker because it is a master regulator of adipogenesis [40]. Osteoblast marker genes were selected due to their implication in osteoblastogenesis. ALPL is crucial to bone mineralization by its implication in phosphate metabolism [41], SPP1 is key in bone mineralisation and osteoblast maturation, and is outstanding in osteoblastogenesis at early stages [42], GPNMB (osteoactivin) is essential to bone formation [43], BMP2 promotes osteoblastogenesis [36] and induces ALPL expression [44], and RUNX2 is a promoter of posts [45]. Consequently, the variation in these gene expressions is expected to result in a modulation of adipogenesis and osteoblastogenesis, respectively.

To test the validity of our lab-made media, we compared our osteoblastic and adipogenic media with commercial media on primary hMSCs. As revealed by their better induction of the expression of differentiation marker genes, we obtained a more affordable, reliable, and efficient differentiation media. Even though commercial media marker gene induction seemed greater, their results were associated with high variability. Notwithstanding, when referring to osteoblastogenic induction, commercial medium induction was null.

When studying the comparison between adipogenic and osteoblastogenic differentiations, glucose concentration could be biased. In order to control this, both lab-made media comprised high-glucose concentrations. A concern about glucose concentrations in osteoblastogenic medium has been previously discussed [23]. Nevertheless, in this work, it was proven that osteoblastogenesis can be fulfilled in the presence of high glucose concentrations. Moreover, this process was even greater in this condition than with commercial media. Supporting this data, it has been previously demonstrated that glucose concentrations did not affect osteoblastogenesis differentiation [23].

Accordingly, these lab-made media, used in combination with standard differentiation protocols, facilitated the detection of the pro-adipogenic commitment of osteoblast precursor cells obtained from aged women. Similarly, they also identified an impaired osteoblastic differentiation of precursor cells, which is consistent with their origin from postmenopausal aged women [46].

Interestingly, these data not only showed the sensitivity of our differentiation method to changes in the osteoblast-adipocyte differentiation balance, but also confirmed that MSCs from aged donors are not an adequate cellular model to work with. Hence, it seems that the use of primary MSCs to study this balance should be limited to MSCs from healthy young donors. However, these cells are scarce and not affordable. As a result, in order to avoid the use of primary MSCs, we searched for an MSC cell line that could be differentiated both to osteoblast and adipocyte. We selected two cell lines, mouse C3H10T1/2 and human hMSC-TERT, optimized their differentiation protocols according to their proliferation rate, and compared their differentiation potential with our improved differentiation media. The number of MSCs lines is not as wide as expected. Among human cells, most of them are MSCs immortalized through TERT, and among mice cell lines, the highest number comes from C3H mice. Therefore, we selected two cell lines, mouse C3H10T1/2 and human hMSC-TERT, with proven capacity to differentiate to either osteoblast [36,47,48,49,50] or adipocyte. In contrast to hMSC-TERT, mouse C3H10T1/2 cells were efficiently differentiated to osteoblast and adipocyte cell fates, as confirmed by the differentiation marker genes. Consistent with this, lipid accumulation in hMSC-TERT differentiated to adipocyte was very limited in comparison to C3H10T1/2.

Accordingly, we suggest that C3H10T1/2 is a better cellular model to study adipocyte-osteoblast differentiation equilibrium. Despite this, it is evident that using a mouse cell line could limit the validity of the data obtained. Nonetheless, we claimed that this cell line might reduce the use of primary hMSCs and that it could be a convenient first approach to study the adipocyte-osteoblast balance, prior to primary cultures from young donors.

In this work, using optimized lab-made media, we provide and prove an easy, affordable, and efficient MSCs differentiation method to study the balance between adipogenesis and osteoblastogenesis. Moreover, we propose an alternative cellular model (C3H10T1/2 cell line) to primary MSCs from young donors that can be efficiently differentiated to osteoblast and adipocyte even better than commonly used human cell lines.

## 5. Conclusions

The described composition of proadipogenic and proosteoblastogenic media in the literature was analyzed in this work. After optimization of these media, using lab-made media, we provided an easy, affordable, and efficient MSCs differentiation method to study the balance between adipogenesis and osteoblastogenesis. Moreover, we proposed an alternative cellular model (C3H10T1/2 cell line) to MSCs that can be efficiently differentiated to osteoblast and adipocyte, and which performed even better than commonly used human cell lines.

## Figures and Tables

**Figure 1 biomedicines-11-00031-f001:**
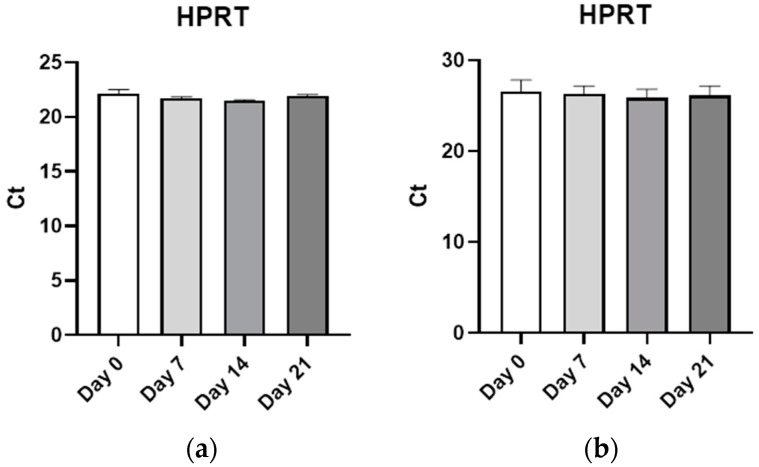
**HPRT expression**. (**a**) HPRT expression along C3H10T1/2 differentiation measured by Rt-qPCR. (**b**) HPRT expression along hMSC-TERT differentiation measured by RT-qPCR.

**Figure 2 biomedicines-11-00031-f002:**
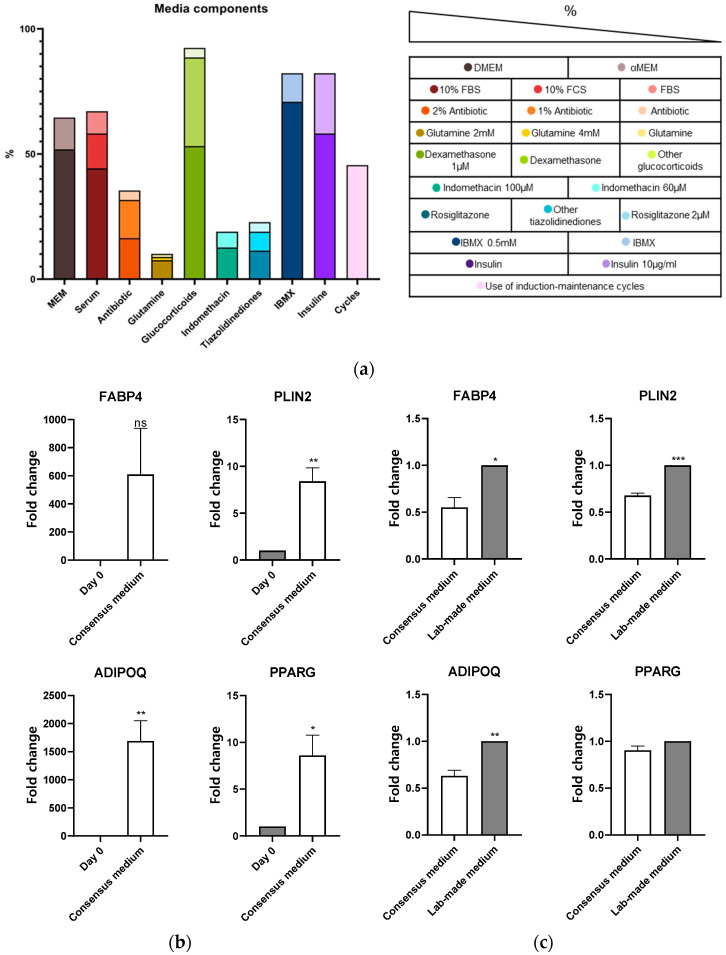
**Design and improvement of a consensus adipogenic medium.** (**a**) Literature analysis of the term “adipogenesis protocol” best results. Percentage of articles using each reagent in the most used concentrations. (**b**) Differentiation of C3H10T1/2 for 7 days with consensus adipogenic medium. FABP4, PLIN2, ADIPOQ, and PPARG marker gene expression assessed by RT-qPCR. Data coming from at least three independent experiments is presented as Fold change over day 0 and expressed as the mean ± SEM (* *p* < 0.05; ** *p* < 0.01; *** *p* < 0.001; **** *p* < 0.00001). (**c**) 7 days C3H10T1/2 adipogenic differentiation with consensus and Lab-made medium. FABP4, PLIN2, ADIPOQ, PPARG marker gene expression assessed by RT-qPCR. Data coming from at least three independent experiments is presented as Fold change over Lab-made medium and expressed as the mean ± SEM (* *p* < 0.05; ** *p* < 0.01; *** *p* < 0.001).

**Figure 3 biomedicines-11-00031-f003:**
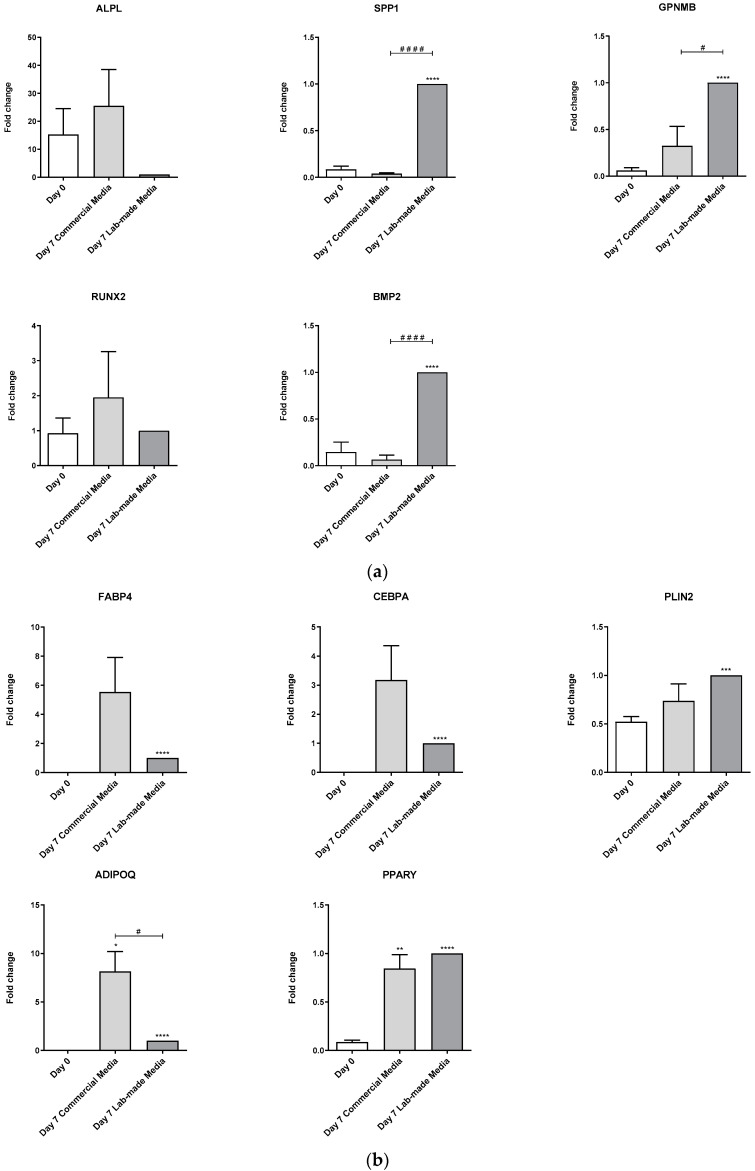
Comparison between commercial and lab-made media adipogenic and osteoblastogenic differentiation induction in primary hMSC. (**a**,**b**) Differentiation of primary hMSC for 7 days with commercial and lab-made media. FABP4, CEBPA, PLIN2, ADIPOQ, and PPARγ adipogenic marker gene expression measured by RT-qPCR. Data coming from at least three independent experiments are presented as fold change over lab-made differentiation data and expressed as the Mean ± SEM (* *p* < 0.05; ** *p* < 0.01; *** *p* < 0.001; **** *p* < 0.00001). Differences between commercial medium and Lab-made medium were indicated as # *p* < 0.05; #### *p* < 0.00001.

**Figure 4 biomedicines-11-00031-f004:**
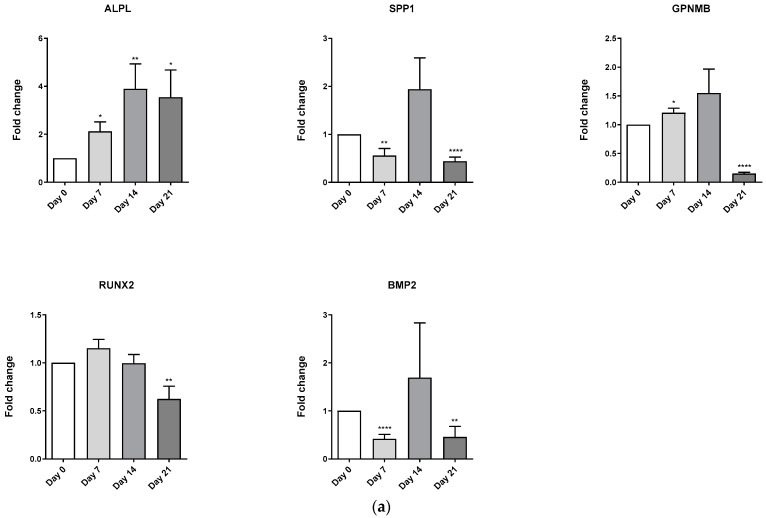
**Adipocytic and osteoblastic differentiation of human pre-osteoblast-like cells.** (**a**) Adipocytic Lab-made medium FABP4, CEBPA, PLIN2, ADIPOQ, and PPARγ gene expression induction in human pre-osteoblast like cells. (**b**) Osteoblastic Lab-made medium ALPL, SPP1, GPNMB, RUNX2 gene expression induction. Data coming from at least three independent experiments (RT-qPCR) is presented as Fold change over D0 differentiation data and expressed as the Mean ± SEM (* *p* < 0.05; ** *p* < 0.01; *** *p* < 0.001; **** *p* < 0.00001).

**Figure 5 biomedicines-11-00031-f005:**
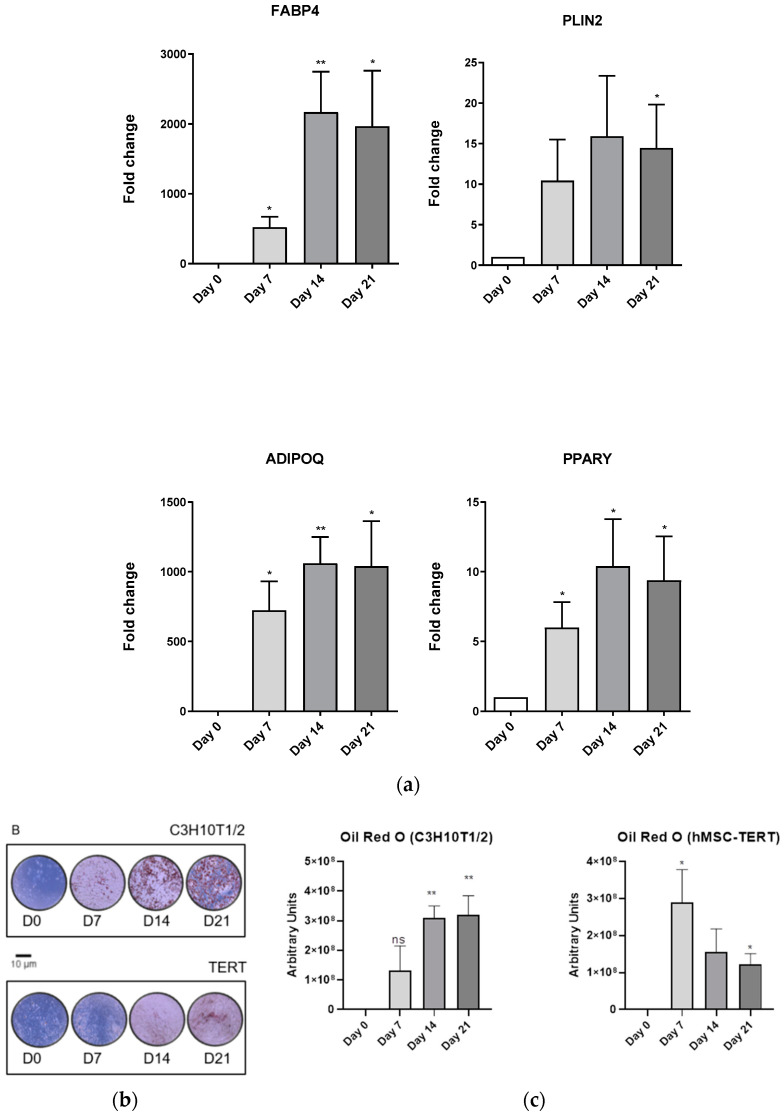
**Adipocytic differentiation of C3H10T1/2, hMSC-TERT cell lines.** (**a**) Differentiation of C3H10T1/2 along 21 days with lab-made medium. (**b**,**c**) Oil Red O staining of C3H10T1/2 and hMSC-TERT cells differentiated through 21 days with lab-made medium. (**d**) Differentiation of hMSC-TERT along 21 days with lab-made medium. FABP4, PLIN2, ADIPOQ, and PPARγ adipogenic marker gene expression assessed by RT-qPCR. Data coming from at least three independent experiments is presented as fold change over D0 differentiation data and expressed as the Mean ± SEM (* *p* < 0.05; ** *p* < 0.01; *** *p* < 0.001).

**Figure 6 biomedicines-11-00031-f006:**
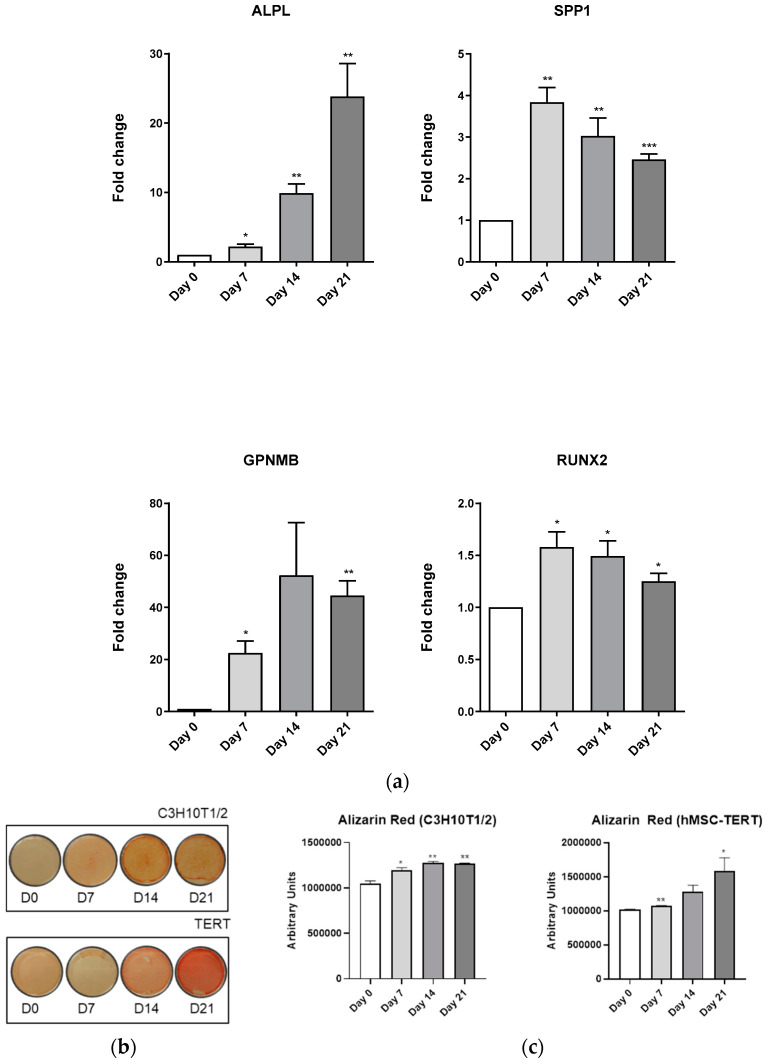
**Osteoblastic differentiation of C3H10T1/2, and hMSC-TERT cell lines.** (**a**) Differentiation of C3H10T1/2 along 21 days with lab-made medium. (**b**,**c**) Alizarin Red staining of C3H10T1/2, and hMSC-TERT cells differentiated during 21 days. (**d**) Differentiation of hMSC-TERT during 21 days with lab-made medium. ALPL, SPP1, GPNMB, and RUNX2 osteoblastogenic marker gene expression assessed by RT-qPCR. Data coming from at least three independent experiments are presented as fold change over D0 differentiation data and expressed as the Mean ± SEM (* *p* < 0.05; ** *p* < 0.01; *** *p* < 0.001; **** *p* < 0.00001).

**Table 1 biomedicines-11-00031-t001:** Primers used for RT-qPCR.

Description	Symbol	Forward Primer (5′-3′) Sequence	Reverse Primer (5′-3′) Sequence
**Human Alkaline phosphatase**	ALPL	TCTTCACATTTGGTGGATAC	ATGGAGACATTCTCTCGTTC
**Human Osteopontin**	SPP1	GACCAAGGAAAACTCACTAC	CTGTTTAACTGGTATGGCAC
**Human Osteoactivin**	GPNMB	CAGATCAGATTCCTGTGTTTG	ACAGTATGATTGGTGGAAAC
**Human RUNT-related transcription factor 2**	RUNX2	AAGCTTGATGACTCTAAACC	TCTGTAATCTGACTCTGTCC
**Human Bone morphogenic protein 2**	BMP2	TCCACCATGAAGAATCTTTG	TAATTCGGTGATGGAAACTG
**Human Fatty acid binding protein 4**	FABP4	CAAGAGCACCATAACCTTAG	CTCGTTTTCTCTTTATGGTGG
**Human CCAAT Enhancer Binding Protein Alpha**	CEBPA	AGCCTTGTTTGTACTGTATG	AAAATGGTGGTTTAGCAGAG
**Human Perilipin 2**	PLIN2	GTTCACCTGATTGAATTTGC	GAGGTAGAGCTTATCCTGAG
**Human Adiponectin**	ADIPOQ	GGTCTTATTGGTCCTAAGGG	GTAGAAGATCTTGGTAAAGCG
**Human Peroxisome proliferator activator receptor γ**	PPARG	AAAGAAGCCAACACTAAACC	TGGTCATTTCGTTAAAGGC
**Human Hypoxanthine Phosphoribosyl transferase 1**	HPRT	ATAAGCCAGACTTTGTTGG	ATAGGACTCCAGATGTTTCC
**Mouse Alkaline phosphatase**	ALPL	ATTCCCACTATGTCTGGAAC	CTCAAAGAGACCTAAGAGGTAG
**Mouse Osteopontin**	SPP1	GGATGAATCTGACGAATCTC	GCATCAGGATACTGTTCATC
**Mouse Osteoactivin**	GPNMB	CTCTTTAATGCCTACTGGTTAC	GCCATATCTGTTTATTCGGC
**Mouse RUNT-related transcription factor 2**	RUNX2	ACAAGGACAGAGTCAGATTAC	CAGTGTCATCATCTGAAATACG
**Mouse Fatty acid binding protein 4**	FABP4	GTAAATGGGGATTTGGTCAC	TATGATGCTCTTCACCTTCC
**Mouse Perilipin 2**	PLIN2	ATAAGCTCTATGTCTCGTGG	GCCTGATCTTGAATGTTCTG
**Mouse Adiponectin**	ADIPOQ	CCACTTTCTCCTCATTTCTG	CTAGCTCTTCAGTTGTAGTAAC
**Mouse Peroxisome proliferator activator receptor γ**	PPARG	AAAGACAACGGACAAATCAC	GGGATATTTTTGGCATACTCTG
**Mouse Hypoxanthine Phosphoribosyl transferase 1**	HPRT	AGGGATTTGAATCACGTTTG	TTTACTGGCAACATCAACAG

## Data Availability

Data available upon reasonable request to the corresponding author.

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
