# Peer review of "Improved Protocol to Study Osteoblast and Adipocyte Differentiation Balance"

_biomedicines, 2022, doi:10.3390/biomedicines11010031_

Round 1

Reviewer 1 Report

In this reviewer opinion, the manuscript is clear, generally well-written and organizated. The statistical analysis should be improve, please add normality data distribution test.

Author Response

We would like to thank the reviewer for their comments. They helped us to improve the manuscript, so we have really appreciated their opinion and suggestions.

In this reviewer opinion, the manuscript is clear, generally well-written and organizated. The statistical analysis should be improve, please add normality data distribution test.

As the reviewer can see, we have taken into consideration all the suggestions raised, and we have modified the article in order to agree with these comments. We hope that our amendments about the normality data distribution tests performed will make this paper deemed fit for publication in Biomedicines. Also, we have rewritten the misspellings found in the text.

Reviewer 2 Report

Authors suggest to have improved a protocol to study osteoblast vs.d adipocyte differentiation”, for which inadequate data is provided.

First off, I do not understand the graph/table on page 6. What am I supposed to extract from this? Standard adipogenic differentiation media typically include insulin, dexamethasone, and IBMX – in highly variable concentrations (e.g., Scott et al. Stem Cells & Development, 2011). Excellent comparisons are also published for the osteogenic counterpart.

Authors state that “current human mesenchymal stem cells in vitro differentiation models are expensive, and hardly reproducible”. Probably, not everybody would agree to this. In light of hundreds of papers with overlapping claims. Specifically, and while their medium composition may slightly diverge from the averaged protocol, I am missing novelty. Their adipogenic medium, for instance, contains about everything anybody else would use, including costly IGF1/insulin. So, where is this less expensive or more reproducible? Authors further state that they are providing an easy, affordable, and efficient MSCs differentiation method. Why would it be more reproducible, easier or efficient, please?

In support of their claims, authors provide mainly “real-time PCR” data of an array of transcription factors. Usually, the respective expression ratio of which relative to each other matters most, but was not analyzed. Importantly, what the authors seem to have done is usually referred to as RT-qPCR, which is a specific type of “real-time PCR” suitable for gene expression studies. This may be worth rephrasing.

To this end, the present Methods section is completely inadequate. Authors must provide some basic information as recommended by the MIQE guidelines as otherwise i) reproducibility and ii) objective evaluation of the suitability of their data interpretation is impossible. Also, a rationale for using Hypoxanthine phosphoribosyl transferase as the sole “housekeeper” must be provided. Of note, a 2018 report on the selection of “housekeepers” for iPSC reprogramming reported Hprt as one of the least suitable reference genes.

Few supposedly “representative” images of histochemical stainings are provided, but were unfortunately not quantified - not densiometrically and not spectrophotometrically - both of which are routine practice in the field. Please add. Thus, conclusions more or less exclusively rely on “real-time PCR” data.

Most importantly, however, in all described cultures of the present study FBS was used, without providing the specific batch No. Popov et al. (IJMS 2019) described the long-known impact of serum in mesenchymal progenitor cell differentiation experiments, explicitly for hMSCs. Serum batch is the single most important variable in stem cell differentiation experiments. Due to its importance, comparing chemically defined differentiation media would be meaningful. Importantly, present authors even used different types of FBS, either being “inactivated”? or not. This, to my mind, questions the comparability of their approach. No commercial entities were ever willing to share their medium components. Aiming at comparing custom-made media to the latter is thus of questionable value.

Notably, authors “propose C3H10T1/2 cells as an alternative model to MSCs that can be efficiently differentiated to osteoblast and adipocyte even better than commonly used human cell lines”. To my mind, this claim exceeds by far what is supported by data as simply a single human cell line (“hMSC-TERT from Dr. Campana) was used as reference. Proposing C3H10T1/2 as a surrogate for human primary MSCs is also questionable as stem cells from different species have completely different in vitro culture requirements during, especially, osteogenic differentiation.

“Every differentiation process was performed for 21 days as long as cells remained attached” – you claim to have improved a protocol to study osteoblast differentiation and you are not sure if cells stayed attached over three weeks?

Reviewer 3 Report

The article entitled Improved protocol to study osteoblast and adipocyte differentiation balance is a document of interesting subject matter.

1. It is expected to have an extensive literature review followed by an in-depth and critical analysis of the state of the art, and identify challenges for future research in the Introduction.

2. The authors should do the analysis the conclusion section must clearly establish a strong correlation with the proposed topic.
3. Your abstract should clearly state the essence of the problem you are addressing, what you did and what you found and recommend. That will help a prospective reader of the abstract to decide if they wish to read the entire article

4. The objective or objectives should be clearly elucidated in the last paragraph of the introduction.

5.         Please pay attention on more interpretation of the experimental results.

Author Response

Reviewer 3:

We would like to thank very much this reviewer for their careful reading of our manuscript and for the chance to improve it. As the reviewer can see, we have considered all of his/her suggestions and comments and corrected all the misspellings and grammar mistakes present in the text. Here you can find point-to-point replies to your observations:

It is expected to have an extensive literature review followed by an in-depth and critical analysis of the state of the art, and identify challenges for future research in the Introduction.

Thank you for this wise comment. We agree with reviewer 3 that is needed an extensive literature review in the introduction, therefore, some key new information was added to it. Nonetheless, the extensive literature review and further analysis were included as a result. Similarly, the challenges for future research were stated in the discussion, once the results obtained allowed us to identify them.

The authors should do the analysis the conclusion section must clearly establish a strong correlation with the proposed topic.

Thank you for this suggestion. As reviewer 3 suggested, we have established, in the conclusion paragraph, the relation with the proposed topic.

Your abstract should clearly state the essence of the problem you are addressing, what you did and what you found and recommend. That will help a prospective reader of the abstract to decide if they wish to read the entire article

Thank you for this comment. We have also considered the suggestions about the abstract information. Even though we are limited by the word count, we focused the information provided as suggested.

The objective or objectives should be clearly elucidated in the last paragraph of the introduction.

Thank you for this suggestion. The introduction was also completed with the description of the objectives of the work, as demanded by reviewer 3.

We hope that with these amendments our paper achieves the priority for publication.

Reviewer 4 Report

The authors compared differentiation Lab-made media with commercial media and tried to identify a cell-line to simultaneously evaluate adipogenic and osteogenic differentiation of MSCs. The manuscript is clear, but sometimes not so easy to understand. The following concerns should be addressed before the manuscript can be considered for publication. 

- The authors need to explain better what do they mean by “adipocyte-osteoblast-equilibrium”

- Mesenchymal stem cells can also differentiate into chondrogenic lineage. Why this was not evaluated?

- Characterization and more information about the MSCs used in this study should be incorporated, such as donor information (age, gender) and flow cytometry analysis.

- The Lab made OB medium is the common medium used in most papers related with osteogenic differentiation, composed mainly by dexamethasone, beta glycerophosphate and ascorbic acid. What is the novelty of the medium used in this study?

- The authors state that “Every differentiation process was performed for 21 days as long as cells remained attached”. Did the cells detach the wells? If so, was the experiment compromised?

- Figure 3 – Were the cells differentiated into an osteogenic lineage for 7 or 14 days? The graphs show 7 days, however the text says 14 days. Furthermore, the legend is incorrect, since the first panel is related with osteogenic differentiation and the second one with adipogenic differentiation.

- Line 273 – Please, correct “cekks”.

- Figure 4 – Why did the cells increase osteogenic gene expression after 14 days of differentiation but decrease after 21 days?

- Why did the authors decide to compare two different MSC lines from different sources (mice and human). These results can not be compared.

- Authors should include other quantitative data instead of quantification of Alizarin Red and Oil Red O stainings. Other analysis, such as calcium content or ALP quantification should be performed.

-Line 350 – How can the authors claim that the media used are affordable? Did they perform any economic analysis comparing these media with commercial ones?

- Line 361 – What do authors mean by “most stable media”?

-The authors supplemented the adipogenic medium with IGF-1 to boost adipogenesis, however for the osteogenic differentiation medium, the authors did not want to use other factors, such as BMP-2, since they did not want to mask the effect of the other components of the medium. The same strategy should have been applied for both differentiation protocols.

- The authors decided to use osteogenic differentiation medium composed by high concentrations of glucose. Although they claim that glucose concentration did not affect osteogenic differentiation, some studies have reported that high glucose medium can affect the proliferation and osteogenic differentiation of MSCs.

Round 2

Reviewer 2 Report

Authors should comply with MIQE guidelines for RT-qPCR and should provide data in support of HPRT being a valid sole calibrator. Table 1 lacks essential information (see MIQE).

Author Response

We would like to thank the reviewer for their comments. They helped us to improve the manuscript, so we have really appreciated their opinion and suggestions. Following, we answered point to point to the suggestions made: 

Authors should comply with MIQE guidelines for RT-qPCR and should provide data in support of HPRT being a valid sole calibrator. Table 1 lacks essential information (see MIQE).

Thank you for this interesting points. Following MIQE guidelines, we completed the material and methods section with the most relevant aspects of RT-qPCR technique employed. We have tried not to write bulky material and methods section so we have selected the most relevant data. 

As the reviewer suggested, we have included in the text data to support the use of HPRT as the sole housekeeping (now Figure 1). 

We hope that with these amendments our paper achieves the priority for publication.

Reviewer 3 Report

Accept.

Author Response

Thank you for your comments

Reviewer 4 Report

The authors addressed most of the reviewer's comments.

Round 3

Reviewer 2 Report

The claim that current hMSCs in vitro differentiation models are expensive or hardly reproducible is not supported by sufficient evidence. Importantly, the present study aim was to compare differentiation in “lab-made media”to “commercial” media. The composition of the latter is proprietary and usually not disclosed. Moreover, a single cell line (C3H10T1/2) and primary cells from one donor (hMSC) were compared. Given the widely accepted variability between MSC sources this design is not in support of the rather bold conclusions. Technical shortcomings further warrant a cautious interpretation of the results.
